# KGQUIZ: Evaluating the Generalization of Encoded Knowledge in Large Language Models

## ABSTRACT

Large language models (LLMs) demonstrate remarkable performance on knowledge-intensive tasks, suggesting that real-world knowledge is encoded in their model parameters. However, besides explorations on a few probing tasks in limited knowledge domains, it is not well understood how to evaluate LLMs' knowledge systematically and how well their knowledge abilities generalize, across a spectrum of knowledge domains and progressively complex task formats. To this end, we propose KGQUIZ, a knowledge-intensive benchmark to comprehensively investigate the knowledge generalization abilities of LLMs. KGQUIZ is a scalable framework constructed from triplet-based knowledge, which covers three knowledge domains and consists of five tasks with increasing complexity: true-or-false, multiple-choice QA, blank filling, factual editing, and open-ended knowledge generation. To gain a better understanding of LLMs' knowledge abilities and their generalization, we evaluate 10 open-source and black-box LLMs on the KGQUIZ benchmark across the five knowledge-intensive tasks and knowledge domains. Extensive experiments demonstrate that LLMs achieve impressive performance in straightforward knowledge QA tasks, while settings and contexts requiring more complex reasoning or employing domain-specific facts still present significant challenges. We envision KGQUIZ as a testbed to analyze such nuanced variations in performance across domains and task formats, and ultimately to understand, evaluate, and improve LLMs' knowledge abilities across a wide spectrum of knowledge domains and tasks.

**ACM Reference Format:**
Anonymous Author(s). 2023. KGQUIZ: Evaluating the Generalization of Encoded Knowledge in Large Language Models. In *Proceedings of Make sure to enter the correct conference title from your rights confirmation emai (Conference acronym 'XX).* ACM, New York, NY, USA, 17 pages. https://doi.org/XXXXXXX.XXXXXXX

## 1 INTRODUCTION

Large language models (LLMs) have demonstrated incredible abilities to encode and represent real-world knowledge in their model parameters, advancing knowledge-intensive tasks such as open-domain question answering [15, 16, 34, 62, 63, 68], dialogue generation [1, 13, 36], summarization [18, 37, 67], and more. However, their knowledge abilities could also be quite brittle, with LLMs generating hallucinated information [3, 8, 24, 39, 46], struggling to encode

long-tail facts [39], and falling short of abstaining when relevant information is not present in model parameters [7].

As a result, studies and benchmarks have been proposed to probe the knowledge abilities of LLMs [12, 21, 41, 48, 55]. Later works also looked into temporality, evaluating whether LLMs could tackle time-sensitive facts and information [12]. In addition to merely probing LLM knowledge, knowledge-intensive tasks such as open-domain QA [28, 32, 47], fact-checking [33, 40, 47], and more are also proposed and employed to evaluate LLM knowledge abilities. Despite these works' contributions to understanding and expanding the stored information of LLMs, we identify two important yet underexplored factors in LLM knowledge abilities.

**Knowledge Utilization**: Previous works have primarily focused on limited task formats such as fill-in-the-blank questions to test the model's knowledge abilities [44, 48, 53]. However, the complexity or format of a task might influence a model's knowledge abilities, while this crucial aspect often goes unaddressed in the current literature. For example, *factual editing* [2, 6] requires the model to identify factual inconsistency and make corrections, rather than simply evaluating memorization; *reasoning with structured knowledge* [9, 64] examines the model's ability to model knowledge in networks and graphs, instead of only probing knowledge at the atomic level. That being said, how well do LLM knowledge abilities generalize to tasks and contexts of varying format and complexity remain underexplored.

**Knowledge Breadth**: Existing works predominantly consider Wikipedia or a specific domain like biomedical knowledge as the knowledge source for evaluation. However, it has been observed that LLM performance can vary significantly across different knowledge domains [41, 55] - an aspect that has not been adequately addressed in the previous works of LLM knowledge probing and understanding. As a result, the lack of a multi-domain knowledge evaluation of large language models, covering diverse knowledge sources, subject areas, and more, is hindering a comprehensive understanding of LLM knowledge abilities.

To this end, we propose KGQUIZ, a comprehensive benchmark designed to evaluate the knowledge abilities of LLMs across multiple knowledge utilization patterns in diverse knowledge domains. Specifically, KGQUIZ is constructed with structured information from knowledge graphs (KGs) from three varying domains, representing commonsense, encyclopedic, and domain-specific (biomedical) knowledge. For each knowledge graph, KGQUIZ presents a collection of 41,000 knowledge-intensive questions, covering five tasks of increasing complexity: *true-or-false*, *multiple choice*, *blank-filling*, *multi-hop factual editing*, and *open-ended text generation*. These progressively difficult tasks represent the multitudes of LLM knowledge and reasoning abilities, providing a comprehensive and comparative setting to assess LLMs' abilities: they respectively test LLMs' abilities to *judge factual correctness*, *select facts based on*

*model confidence*, *retrieve entities*, *perform factual editing*, and *generate long-form knowledge documents*, presenting a holistic probe of LLM knowledge abilities in different application scenarios.

We evaluate 10 open-source and black-box LLMs on the KGQUIZ benchmark to better understand which LLM covers what knowledge domain better, and under which utilization contexts. Our experiments demonstrate that: 1) **LLM performance greatly varies across knowledge domains.** For instance, on *Task 5: Open-Ended Text Generation*, ChatGPT [45], ChatGLM [14], and TEXT-DAVINCI-003 [45] respectively perform best when it comes to YAGO, ConceptNet, and UMLS, three knowledge graphs representing varying knowledge domains. 2) **Knowledge utilization greatly impacts LLM's ability to retrieve and employ factual knowledge.** For instance, ChatGPT's performance on biomedical knowledge drops by 30% from the fill-in-the-blank task to the factual editing task, suggesting that the additional multi-hop context in factual editing poses new challenges to LLM knowledge abilities. Together, our extensive experiments demonstrate that probing the knowledge abilities of LLMs is nuanced and multi-faceted, with the largest LLMs excelling in simple knowledge utilization tasks on general knowledge domains, while advanced knowledge contexts and domain-specific information remain open challenges. We envision KGQUIZ as a valuable testbed to understand, evaluate, and improve LLM knowledge abilities across varying knowledge domains and utilization contexts.

## 2 THE KGQUIZ BENCHMARK

KGQUIZ employs knowledge graphs from diverse domains to construct five knowledge-intensive tasks with increasing complexity. We denote a knowledge graph as a set of triples $\mathcal{T}$, where the $k$-th triple is $\mathcal{T}_k = (h_k, r_k, t_k)$, and $h_k$, $r_k$ and $t_k$ represent the head entity, relation, and tail entity, respectively. We use $\mathcal{E}$ and $\mathcal{R}$ to denote the sets of all entities and relations in the knowledge graph.

### 2.1 *Task 1: True-or-False*

As a base assessment of knowledge abilities, True-or-False questions ask whether a given statement is factually correct or not. In a way, this task tests the LLMs' ability to verify the factuality of KG-based information, which is the most fundamental ability to distinguish between true and false knowledge [10].

**Task Formulation** We construct two sets of KG triples to represent positive and negative samples ($\mathcal{T}_{pos}$ and $\mathcal{T}_{neg}$). For a positive triple $(h, r, t) \in \mathcal{T}_{pos}$, we replace the tail entity $t$ with another entity $t'$ to generate a negative sample and add it to $\mathcal{T}_{neg}$. We then use the prompt for the positive or negative triple $(h, r, t)$: "*Is the statement h r t True or False?*". We expect LLMs to answer with *True* or *False*, indicating their judgment of the knowledge statement based on their parametric knowledge.

**Negative Sampling** We propose four approaches to sample negative entities $t'$ in the knowledge graph to obtain increasingly challenging negative samples.

- **Random** We randomly sample an entity from a set of entities not connected to the head entity $h$ as $t'$, formally $t' \in \mathcal{E} - \mathcal{E}(h)$, where $\mathcal{E}(h)$ denotes the set of entities connected to $h$.

- **Semantic Similarity** We hypothesize that semantically similar entities could provide a more challenging setting with harder negative examples. We first use the **Random** method to sample $m$ negative entities. These sampled entities form the set $\mathcal{E}_m$. Then, we employ an encoder-based language model, denoted as $\text{enc}(\cdot)$, to encode the names of these entities. Finally, we use cosine similarity $\text{sim}(\cdot, \cdot)$ to select an entity $t'$ that is most similar to $t$ in the embedding space. Formally, $t' = \text{argmax}_{e \in \mathcal{E}_m} \text{sim}(\text{enc}(e), \text{enc}(t))$.

- **Relation Sharing** We hypothesize that using entities sharing the same relation, $r$, as the selected negative sample would provide a challenging adversarial setting. We first obtain the set of entities that are also associated with relation $r$ as $\mathcal{E}^{(r)}$, then randomly sample one entity from $\mathcal{E}^{(r)}$ as the negative sample $t'$.

- **Network Proximity** We hypothesize that entities that are close to $h$ in the KG could also present a hard negative example. We obtain the set of entities that are connected to $h$ and randomly sample one entity from it as the negative sample $t'$.

**Evaluation** We use accuracy as the evaluation metric for the binary output of *True* or *False*.

### 2.2 *Task 2: Multiple-Choice*

Building up from the True-or-False task, the multiple-choice task introduces distractors [22, 50, 56]. This task not only tests the ability of LLMs to determine what is factually correct, but also their ability to discern the false options from the true option. Therefore, the Multiple-choice task presents a higher degree of complexity, as LLMs need to evaluate the plausibility of different answer options based on their parametric knowledge.

**Task Formulation** We randomly sample a subset of the knowledge graph, denoted as $\mathcal{T}_s$. For $(h, r, t) \in \mathcal{T}_s$, we replace the tail entity $t$ with *[MASK]* and provide $m$ answer options, including the correct entity $t$ and $m - 1$ distractors. We follow the same negative sampling strategies in *Task 1: True-or-False* to obtain the distractors.

**Evaluation** We similarly use accuracy as the evaluation metric.

### 2.3 *Task 3: Blank-Filling*

The Blank-filling task requires LLMs to directly generate the missing information for a given statement [48], compared to the two previous tasks where the correct answer already appeared somewhere in the prompt context. While in tasks 1 and 2, models might just take guesses as they can simply choose one of the available options without knowing the actual answer, in *Task 3: Blank-Filling*, LLMs are required to retrieve the correct answer without any hints or options.

**Task Formulation** We randomly sample one subset of the knowledge graph, denoted as $\mathcal{T}_s$. For $(h, r, t) \in \mathcal{T}_s$, we replace the tail entity $t$ with *[MASK]*. The model is asked to generate the correct answer to replace *[MASK]*.

**Evaluation** We denote the model output as $t_o$ and we use the following metrics for evaluation:

- **LCS**: We denote the Longest Common Subsequence of $t_o$ and $t$ as $s$, and LCS is defined as: $\text{LCS} = \frac{\text{Len}(s)}{\max\{\text{Len}(t_o), \text{Len}(t)\}}$

- **F1-score**: We denote the set of common tokens in both $t_o$ and $t$ as $C$. We denote the F1-score of $t_o$ and $t$ as $\text{F1} = \frac{2PR}{P+R}$, where $P = \frac{|C|}{|t_o|}, R = \frac{|C|}{|t_g|}$.

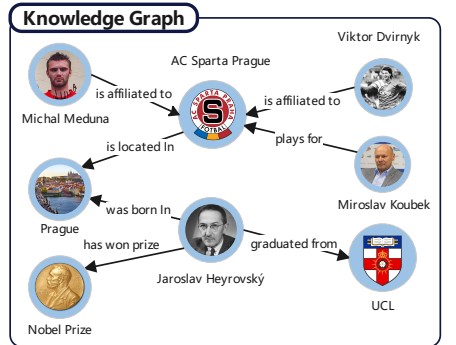

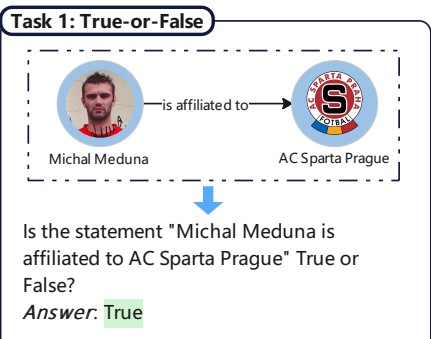

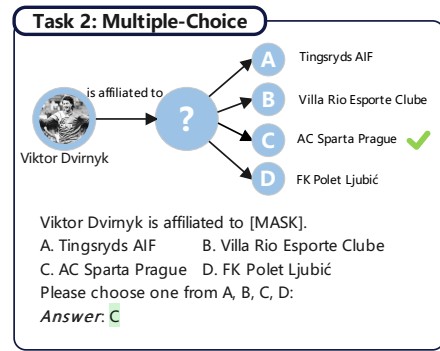

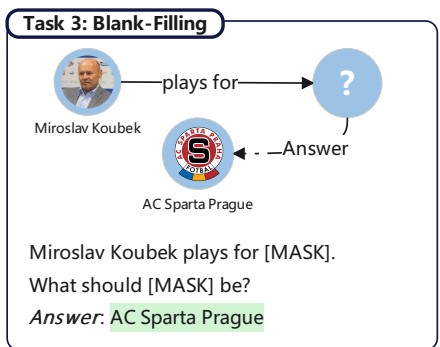

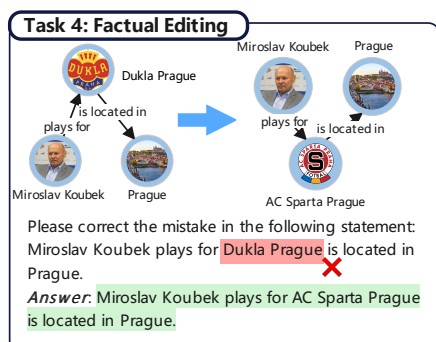

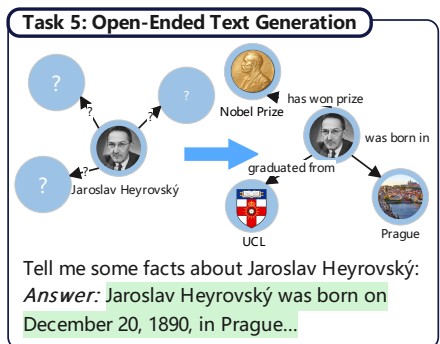

**Figure 1: Overview of the KGQUIZ Benchmark, featuring five knowledge-intensive tasks with increasing complexity. We illustrate the diverse tasks employed in KGQUIZ to test large language models, highlighting the examples and corresponding natural language prompts used to examine their knowledge abilities across domains and contexts.**

- **Semantic Match**: We measure semantic similarity between the model's output and the correct answer using cosine similarity on embeddings obtained via InstructGPT Ada LLM enc(·). This gives us the $\text{AdaScore}(t_o, t) = \text{sim}(\text{enc}(t_o), \text{enc}(t))$. A threshold $\theta$ of Adascore is based on a held-out validation set (detailed in Appendix D) to determine whether the model-generated answer and the ground truth are a semantically exact match. Concretely, we define the semantic match metric as $\text{SM}(t_o, t) = 1$ if $\text{AdaScore}(t_o, t) \geq \theta$, else 0.

## 2.4 Task 4: Factual Editing

The Factual Editing task presents enhanced challenges compared to task 3 by moving from a single knowledge statement to a multi-hop knowledge statement. Task 4 requires LLMs to not only memorize and recall the facts, but also to identify which part of multi-hop knowledge is inconsistent and revise accordingly. While previous works have also explored LLMs' potential in factual editing [2, 6], we uniquely focus on a multi-hop format where one of the hops features inconsistent factual information. This task tests LLMs' abilities to handle multi-hop information, localize errors, edit factual inconsistencies, and more.

**Task Formulation** Given a knowledge graph, we first sample a $k$-hop path, and we use a structured format to present the multi-hop knowledge path as $\boldsymbol{d} = (h_1, r_1, e_1, r_2, ..., t_k)$.[1] We then randomly

replace one of the entities in the path (denoted as $e_s$) with $e'$ sampled with the negative sampling strategies described in Section 5 to obtain $\boldsymbol{d}'$. We concatenate the names of original entities and relations to form a multi-hop knowledge statement denoted as $\boldsymbol{d}$ and swap one entity with its negative sample to obtain $\boldsymbol{d}'$. This task prompts LLMs to correct the factual inconsistency in $\boldsymbol{d}'$.

**Evaluation** We denote the left part of $\boldsymbol{d}$ (tokens before $\epsilon(e_s)$) as $\boldsymbol{L}$, and the right part of $\boldsymbol{d}$ (tokens after $\epsilon(e_s)$) as $\boldsymbol{R}$. We first perform the longest common substring match between the output $\boldsymbol{d}^{(o)}$ of the model and $\boldsymbol{L}$, $\boldsymbol{R}$ in turn, and delete the obtained common substring from $\boldsymbol{d}^{(o)}$ to retrieve the revised entity given by LLMs. Then, We adopt the same set of evaluation metrics as task 3, namely LCS, F1-SCORE, and SEMANTIC MATCH, to compare the ground truth entity $e_s$ and the revised entity given by LLMs.

## 2.5 Task 5: Open-Ended Text Generation

The Open-Ended Text Generation task moves from handling isolated facts (as in the previous tasks) to generating multiple factual associations about a given entity. We evaluate whether the generated factual associations are aligned with the information in existing knowledge graphs. This comparison aims to measure the ability of LLMs to generate accurate and comprehensive factual knowledge of a particular entity. In addition, while tasks in previous works mostly focus on a single factual association [22, 56], we propose the Open-Ended Text Generation task to encourage the knowledge abilities of LLMs in multi-fact and knowledge synthesis settings.

---

[1]To avoid confusion, we denote $e_m$ as the tail entity $t_m$ of the $m$-th triple in the knowledge path. At the same time, it also serves as the head entity $h_{m+1}$ of the $(m+1)$-th triple in the knowledge path.

**Task Formulation** We randomly sample one subset of KG, denoted as $\mathcal{T}_s$. For $(h, r, t) \in \mathcal{T}_s$, we ask the model to *"Tell me some facts about h"*. We denote all triplets containing $h$ in the knowledge graph as $\mathcal{G} = \{(h, r_g, t_g) \in \mathcal{T}\}$.

**Evaluation** We evaluate Open-Ended Text Generation generation by comparing the model outputs with the information about entity $h$ in the original knowledge graph, denoted as $\mathcal{G}$. Concretely, we first prompt a GPT-3.5 LLM to turn the given model output in natural language into a list of fact triplets $O = \{(h, r_o, t_o)\}$ inspired by previous works [26, 43], where we further evaluate this approach in Appendix D. We then employ the semantic match metric SM in task 3, we define the Precision and Recall between model predictions $O$ and ground truth $\mathcal{G}$ as: Precision $= \frac{|O \cap \mathcal{G}|}{|O|}$, Recall $= \frac{|O \cap \mathcal{G}|}{|\mathcal{G}|}$, where $O \cap \mathcal{G}$ denotes the set of triples that are both in model predictions and the knowledge graph with SM = 1.

## 3 EXPERIMENT SETTINGS

*Knowledge Domains.* In our experiments, we posit that the performance of LLMs in knowledge-intensive tasks is greatly influenced by diverse knowledge domains. Thus, we consider knowledge graphs from three distinct domains in our experiments: commonsense, encyclopedic, and domain-specific. For commonsense knowledge, we leverage the ConceptNet knowledge graph [52] with 1,103,036 entities, 47 relations, and 3,098,674 triples. For encyclopedic knowledge, we adopt the YAGO knowledge graph [38] with 123,182 entities, 37 relations, and 1,089,040 triples. For domain-specific knowledge, we mainly consider the biomedical domain and adopt the UMLS knowledge graph [4] with 297,554 entities, 98 relations, and 1,212,586 triples. By conducting our evaluations across knowledge graphs that span varying domains, we aim to provide a comprehensive assessment of how the knowledge abilities of LLMs fare across diverse knowledge domains.

*Models and Settings.* We evaluate both black-box and open-source LLMs on the KGQUIZ benchmark. For black-box LLMs, we adopt InstructGPT [45] (TEXT-ADA-001, TEXT-BABAGGE-001, TEXT-CURIE-001, and TEXT-DAVINCI-003) and ChatGPT (GPT-3.5-TURBO) through the OpenAI API. For open-source LLMs, we adopt GPT-J [60], OPT (6.7B) [66], ChatGLM [14], LLAMA (7B) [58], and Alpaca [57] in the experiments. We use a temperature of $\tau$ = 0 to reduce randomness.

*Task Settings.* For *Task 1: True-or-False*, we construct 10k examples for each knowledge graph and adopt semantic similarity as the default negative sampling method. In our experiments, we noticed that some LLMs could not answer true-or-false questions based on zero-shot instructions, thus we have added one in-context example to demonstrate the QA format. For *Task 2: Multiple-Choice*, we use four answer options as the default setting and construct 10k examples for each knowledge graph. Here, too, we incorporate a single in-context example for clarification. For *Task 3: Blank-Filling*, we randomly sample 10k triplets for each knowledge graph to generate the blank-filling questions. Moving on to *Task 4: Factual Editing*, we construct 10k knowledge walks for each knowledge graph with the default walk length $k$ = 3. Given that some LLMs struggled with this task, an in-context example is provided. Lastly, for *Task 5: Open-Ended Text Generation*, we select 1k entities in each knowledge

| Model | Task | | | | | Domain | | | Avg. |
|---|---|---|---|---|---|---|---|---|---|
| | T1 | T2 | T3 | T4 | T5 | YAGO | CPNet | UMLS | |
| ADA | 8.3 | 9.7 | 6.1 | 5.1 | 4.8 | †6.5 | 6.8 | 7.1 | 6.5 |
| BABBAGE | 7.0 | 6.0 | 5.0 | 5.0 | 3.8 | 5.7 | 5.5 | †4.8 | 5.7 |
| CURIE | 8.7 | 9.3 | 2.8 | 4.0 | **2.7** | †5.2 | 6.1 | 5.2 | 5.2 |
| DAVINCI | 2.0 | 2.0 | **1.7** | **1.6** | 3.0 | †**1.9** | **2.0** | **2.3** | **1.9** |
| TURBO | **1.0** | **1.0** | 3.0 | 3.9 | 2.8 | †2.3 | 2.4 | 2.3 | 2.3 |
| GPT-J | 7.0 | 7.3 | 8.7 | 7.7 | 9.0 | 8.0 | †7.6 | 8.1 | 8.0 |
| OPT | 9.0 | 7.0 | 7.0 | 7.8 | 9.8 | †8.2 | 8.5 | 8.3 | 8.2 |
| CHATGLM | 4.7 | 3.0 | 4.0 | 7.1 | 3.8 | 4.3 | †4.0 | 5.3 | 4.3 |
| LLAMA | 4.0 | 5.7 | 8.9 | 8.1 | 7.3 | 7.2 | 7.1 | †6.1 | 7.2 |
| ALPACA | 3.3 | 4.0 | 6.9 | 4.8 | 7.8 | 5.6 | †4.9 | 5.6 | 5.6 |

**Table 1: Overall average rankings of ten LLMs on KGQUIZ across five tasks and three knowledge domains. Bold, underline represents the highest and the second highest ranking on each task (or knowledge domain). † denotes the knowledge domain on which each model has its best ranking.**

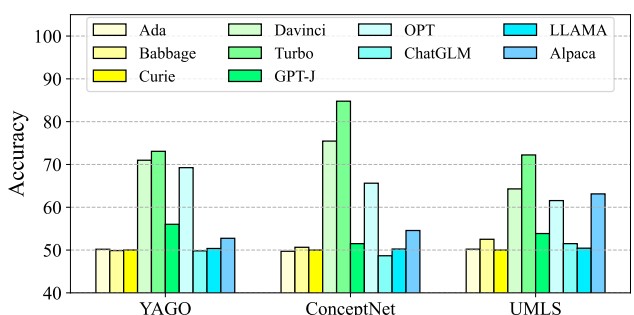

**Figure 2: Model performance on *Task 1: True-or-False*. Larger LMs are better at judging factual correctness, while the same LM performs differently across varying knowledge domains.**

graph and ask LLMs to perform open-ended generation[2]. We use *Semantic Similarity* to sample negative examples in our subsequent experiments.[3]

## 4 RESULTS

We first present the average ranking across the five knowledge reasoning tasks and the three knowledge domains in Table 1. In terms of knowledge domains, we observe a considerable discrepancy in the performances across different domains for the same LLM. This finding highlights that LLM knowledge abilities are greatly impacted by knowledge domain, supporting the need for multi-domain knowledge probing benchmarks such as KGQUIZ. Regarding knowledge utilization, the format in which knowledge is presented and required to be utilized by LLMs also significantly impacts their overall performance, as the best model across the five tasks could be quite different. We further analyze each individual task in the following.

---

[2]For some tasks, we use in-context examples. More details in Appendix D.
[3]The specific effect of these four strategies and our choice for *Semantic Similarity* is detailed in section 5.1.

| Model | YAGO | | | ConceptNet | | | UMLS | | |
|---|---|---|---|---|---|---|---|---|---|
| | **F1-score** | **LCS** | **Sem. Match** | **F1-score** | **LCS** | **Sem. Match** | **F1-score** | **LCS** | **Sem. Match** |
| ADA | 2.26 | 18.24 | 61.67 | 1.24 | 11.76 | 45.43 | 5.72 | 19.43 | 55.52 |
| BABBAGE | 2.60 | 17.63 | 60.48 | 2.07 | 12.06 | 64.67 | 10.37 | 21.68 | 71.43 |
| CURIE | 5.38 | 19.63 | 71.54 | 3.32 | 15.11 | 78.68 | 10.90 | 26.04 | 84.70 |
| DAVINCI | **14.02** | **28.65** | **73.00** | **6.27** | **27.40** | **91.19** | 8.28 | 23.81 | 87.88 |
| TURBO | 4.47 | 11.83 | 52.33 | 5.56 | 14.42 | 80.48 | **19.44** | **28.18** | **89.27** |
| GPT-J | 0.56 | 10.75 | 24.55 | 1.20 | 4.53 | 39.07 | 9.38 | 11.74 | 73.17 |
| OPT | 0.66 | 10.75 | 27.33 | 0.75 | 4.40 | 45.55 | 6.88 | 11.21 | 73.52 |
| CHATGLM | 3.53 | 21.50 | 72.27 | 2.35 | 20.15 | 88.07 | 4.04 | 19.45 | 58.71 |
| LLAMA | 1.24 | 11.43 | 35.97 | 1.03 | 3.42 | 25.96 | 7.44 | 9.31 | 76.64 |
| ALPACA | 3.16 | 10.37 | 41.52 | 1.92 | 6.25 | 56.55 | 10.63 | 13.61 | 81.88 |

Table 2: LLM performance on *Task 3: Blank-Filling*. Sem. Match is short for the semantic match metric. DAVINCI leads on YAGO and ConceptNet, while TURBO performs best on UMLS, indicating that LLM knowledge abilities vary greatly across knowledge domains.

Figure 3: LLM performance on *Task 2: Multiple-Choice*. DAVINCI and TURBO consistently outperform other models, indicating their superior knowledge abilities under the multiple-choice knowledge utilization format.

## 4.1 *Task 1: True-or-False*

As depicted in Figure 2, among the assessed LLMs, four of them (TEXT-DAVINCI-003, GPT-3.5-TURBO, ChatGLM) performed substantially better than random chance (50%) on all KGs. Notably, GPT-3.5-TURBO achieved the best overall performance, showcasing its ability to discern correct from incorrect knowledge statements. Observation of improved performance with larger model sizes suggests that models with more parameters can encode more knowledge and leverage the stored knowledge to accurately identify the veracity of knowledge statements. Additionally, Even in the simple binary task, many LLMs show accuracy close to 50%, indicating difficulty in distinguishing true and false statements. This suggests a need for further improvement in LLMs' knowledge abilities, particularly for smaller language models.

## 4.2 *Task 2: Multiple-Choice*

Figure 3 showcases that TEXT-DAVINCI-003 and GPT-3.5-TURBO consistently outperform other LLMs in understanding and applying knowledge across all KGs and domains. An observation from

tasks comparison revealed that TEXT-DAVINCI-003 and GPT-3.5-TURBO's improved performance in *Task 2: Multiple-Choice* compared to *Task 1: True-or-False*. However, Alpaca's relative performance dwindled in Task 2, suggesting that the specific knowledge utilization format significantly influences an LLM's ability to retrieve potentially correct answers.

## 4.3 *Task 3: Blank-Filling*

Compared to true-or-false and multiple-choice questions, blank filling requires LLMs to retrieve the correct answer from their parametric knowledge without relying on any options. In Table 2, the overall low LCS scores reflect that LLMs' generated answers struggle to match the exact target answer. Moreover, the models' abilities differ significantly, with TEXT-DAVINCI-003 excelling in two domains (YAGO and ConceptNet) but GPT-3.5-TURBO performing better in the biomedical domain (UMLS). Additionally, we observe a noticeable decrease in performance in the biomedical domain, suggesting that the models may not be as proficient in handling domain-specific knowledge.

## 4.4 *Task 4: Factual Editing*

Compared to blank-filling, *Task 4: Factual Editing* involves identifying and rectifying factual inconsistencies within given knowledge statements. According to the results in Table 3, the additional context indeed aids certain models in generating fact-checked responses on certain KGs (YAGO and ConceptNet), with TEXT-DAVINCI-003 and GPT-3.5-TURBO scoring well for YAGO and ConceptNet respectively, and ChatGLM excelling on UMLS. It highlights that tasks such as dialogue generation and summarization, which usually come with relevant context, may work better with LLMs. However, when provided only with a short question, QA models may get confused easily. The task-wise change in top-performing models indicates that the form of knowledge utilization impacts an LLM's knowledge abilities significantly.

## 4.5 *Task 5: Open-Ended Text Generation*

Open-ended generation tasks present a more complex challenge to LLMs as it requires not just specific factual associations, but

| Model | YAGO | | | ConceptNet | | | UMLS | | |
|---|---|---|---|---|---|---|---|---|---|
| | F1-score | LCS | Sem. Match | F1-score | LCS | Sem. Match | F1-score | LCS | Sem. Match |
| ADA | 2.50 | 14.51 | 86.76 | 0.12 | 14.65 | 83.84 | 2.50 | 18.11 | 59.85 |
| BABBAGE | 2.90 | 9.47 | 90.68 | 0.02 | 10.42 | 86.53 | 2.90 | 17.78 | 60.03 |
| CURIE | 6.21 | 8.93 | 91.20 | 0.10 | 15.92 | 83.14 | **6.21** | **19.76** | 60.24 |
| DAVINCI | **16.99** | **20.58** | **91.77** | **5.15** | **17.31** | 93.25 | 5.44 | 7.28 | 64.19 |
| TURBO | 12.29 | 13.24 | 91.06 | 0.51 | 1.28 | **93.32** | 0.88 | 8.93 | 59.05 |
| GPT-J | 0.03 | 0.17 | 90.34 | 0.00 | 0.22 | 93.21 | 0.20 | 0.71 | 59.98 |
| OPT | 0.01 | 0.06 | 90.37 | 0.00 | 0.06 | 93.24 | 0.30 | 0.88 | 59.96 |
| CHATGLM | 4.94 | 1.32 | 89.66 | 0.14 | 4.57 | 90.62 | 0.42 | 2.58 | **76.26** |
| LLAMA | 0.03 | 0.04 | 90.33 | 0.00 | 0.00 | 93.20 | 0.43 | 1.81 | 59.98 |
| ALPACA | 6.80 | 12.27 | 90.20 | 0.87 | 14.84 | 93.20 | 1.46 | 8.66 | 59.93 |

**Table 3: LLM performance on *Task 4: Factual Editing*. Model performance is generally higher than blank-filling, indicating the helpfulness of additional context and emphasizing the influence of knowledge utilization. Models such as TURBO, DAVINCI, and ChatGLM show variations in performance across different knowledge graphs, highlighting the influence of knowledge domains.**

| Model | YAGO | | ConceptNet | | UMLS | |
|---|---|---|---|---|---|---|
| | Precision | Recall | Precision | Recall | Precision | Recall |
| ADA | 75.84 | 34.89 | 90.93 | 24.90 | 59.45 | 19.47 |
| BABBAGE | 84.66 | 35.34 | 95.01 | 18.84 | 81.52 | 22.93 |
| CURIE | **85.69** | 38.64 | **96.59** | 22.46 | **83.43** | 26.80 |
| DAVINCI | 76.39 | 53.96 | 88.12 | 41.55 | 77.48 | **46.06** |
| TURBO | 77.28 | **57.63** | 89.39 | 40.53 | 75.94 | 43.89 |
| GPT-J | 11.97 | 8.78 | 24.11 | 12.07 | 10.72 | 5.96 |
| OPT | 14.06 | 7.72 | 16.89 | 5.26 | 10.35 | 5.43 |
| CHATGLM | 71.00 | 54.54 | 88.05 | 46.49 | 63.59 | 39.72 |
| LLAMA | 39.17 | 29.29 | 36.78 | 11.78 | 26.14 | 11.85 |
| ALPACA | 22.96 | 17.77 | 28.63 | 13.94 | 12.69 | 7.53 |

**Table 4: Model performance on *Task 5: Open-Ended Text Generation*. Different from previous tasks, generating long and open-ended statements about entities poses new challenges to LLMs.**

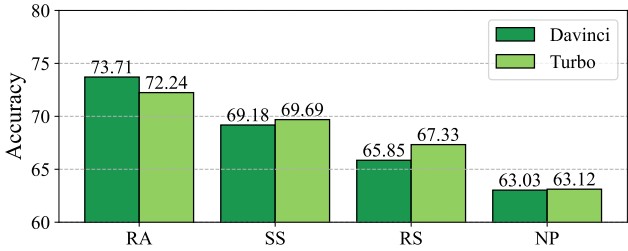

**Figure 4: Performance on *Task 1: Ture-or-False* with varying negative sampling methods. The figure illustrates the performance of TEXT-DAVINCI-003 and GPT-3.5-TURBO on the YAGO knowledge graph when using the four negative sampling strategies, showing that the choice of negative sampling has a significant impact on the difficulty of the task.**

also the generation of a consistent paragraph about a certain entity encapsulating assorted facts and knowledge. As observed in Table 4, TEXT-DAVINCI-003 tops the chart with the highest AdaScore_s score across all three KGs, denoting its proficient ability to produce well-structured and factually accurate knowledge paragraphs. TEXT-CURIE-001 stands out with the highest Precision score, indicating its preference to generate knowledge closely in line with the respective knowledge graph. From a Recall perspective, the best performances are achieved by GPT-3.5-TURBO, ChatGLM, and TEXT-DAVINCI-003 on the three respective KGs. These findings emphasize that the knowledge domain significantly affects the performance of LLMs in knowledge-intensive tasks, underscoring the need for comprehensive evaluations of LLMs' knowledge abilities that consider varying knowledge domains.

## 5 ANALYSIS

### 5.1 Negative Sampling Strategy

In section 2.1, we propose and formalize four negative sampling methods to generated questions in the KGQUIZ benchmark. In order to investigate their impact on the difficulty of the task, we use the

four negative sampling strategies, *Random* (RA), *Semantic Similarity* (SS) *Relation Sharing* (RS), and *Network Proximity* (NP) to generate questions for *Task 1: True-or-False* based on the YAGO knowledge graph. We evaluate TEXT-DAVINCI-003 and GPT-3.5-TURBO as shown in Figure 4. These results show that different negative sampling methods *do* impact on the difficulty of the problem, ranging from easy to difficult in the following order: *Random*, *Semantic Similarity*, *Relation Sharing*, and *Network Proximity*. It is also demonstrated that whether LLMs can select the correct answer is impacted by the plausibility of negative examples.

In particular, we employed *Semantic Similarity* as an intermediate strategy presenting reasonable complexity. This strategy, while challenging, does not make the task excessively difficult. Furthermore, while we propose this specific strategy, KGQUIZ benchmark supports the flexibility of adopting other negative sampling settings.

### 5.2 Consistency Study

In this study, we investigate the robustness towards minor changes in prompts and knowledge statements. We select 100 questions from the YAGO knowledge graph in *Task 1: True-or-False* and evaluate

| Question | Prediction | Gold |
|---|---|---|
| Bob Hawke graduated from ____ | Oxford University | University of Oxford |
| Rosemary Sutcliff has won prize ____ | The Carnegie Medal | Carnegie Medal (literary award) |
| Taito Corporation is located in ____ | Tokyo, Japan | Shibuya, Tokyo |

**Table 5: Qualitative analysis of *Task 3: Blank-Filling*, suggesting that our proposed *Semantic Match* presents a more nuanced metric for knowledge probing.**

with five different prompts and instructions (more details in Appendix E.3). We measure response consistency of the five black-box LLMs using the Fleiss Kappa measure [17]. The experiment results show that LLMs have varying robustness towards prompt formats: TURBO (0.645) has the highest score, suggesting a moderate level of agreement. DAVINCI (0.285) exhibits a lower but still positive value. However, ADA (-0.187), BABBAGE (-0.057), and CURIE (-0.168) show negative Fleiss Kappa values, indicating poor agreement and suggesting that model responses are less consistent towards minor changes in knowledge probing instructions. This study highlights that the robustness to minor changes in knowledge-intensive prompts is in itself part of LLM's knowledge abilities.

## 5.3 Exact Match vs. Semantic Match

We conduct qualitative analysis for *Task 3: Blank-Filling* and present a few examples in Table 5. It is demonstrated that answers generated by LLMs do not exactly match the gold label, where the exact match (EM) metric would treat the answer as incorrect. However, the generated responses are semantically equivalent. For instance, in the first example, the word order is different but both answers convey the same meaning. Similarly, in the third example, "Tokyo, Japan" is more general than the gold answer "Shibuya, Tokyo" but it still provides the correct location information. While the exact match metric would treat them as incorrect, under our proposed *Semantic Match*, all four answers are deemed as correct, indicating that *Semantic Match* presents a better evaluation metric in LLM knowledge probing given the nuanced nature of entity names [31].

## 5.4 Question Sampling

In KGQUIZ, for each task, we generate questions by randomly sampling triplets (or head entities) from the KG, while whether the randomly sampled subsets is represented of the whole KG remain underexplored. To this end, we design two additional ways to sample a problem subset:

- **Relation Proportion**: We first calculate the proportion of relations in the KG, then sample triplets based on the relation distribution. This ensures that the proportion of relations in the sampled triples is consistent with the proportion of relations in the entire knowledge graph.
- **Entity Clustering**: First, we use knowledge graph embedding model TransE [5] to obtain the embedding for each entity, then we

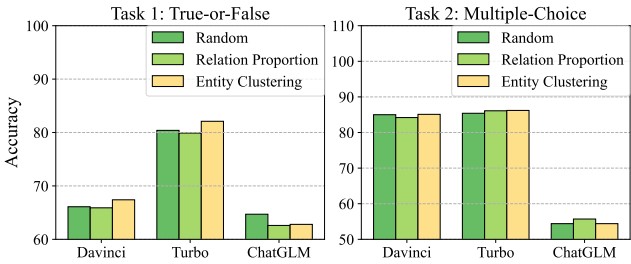

**Figure 5: Comparison of model performance across different question sampling methods. Models are evaluated on 1,000 *Task 1: True-or-False* questions and 1,000 *Task 2: Multiple-Choice* questions sampled via three different methods. The results show the model's performance is not significantly affected by the sampling method.**

use K-means to obtain 10 clusters of entities. We sample triplets based on the proportions of the number of entities in each cluster.

We generated 1,000 *Task 1: True-or-False* questions and 1,000 *Task 2: Multiple-Choice* questions on ConceptNet using these two methods respectively. According to Figure 5, we find that after changing to these two sampling methods that can theoretically better represent the features of the knowledge graph, the performance of each model did not change significantly (compared to random sampling). This indicates that randomly sampled triples can also reflect the features of the entire knowledge graph and the corresponding results are representative.

## 5.5 Negative Sampling Evaluation

*Validity of Negative Samples.* Regarding the four negative sampling methods we proposed, a potential issue is that the sampled data may not be genuine negative samples. Therefore, in order to investigate the effectiveness of our negative sampling methods, we manually evaluated 20 samples for each method. In our manual evaluation, all the sampled examples were indeed true negative samples, which validated the effectiveness of our negative sampling methods.

## 5.6 Number of Options

Although extra answer options could serve as context information aid LLMs (as we analyzed in Section 4.2, we hypothesize that an increasing amount of distractors might sway LLMs away from the correct answer. To this end, we study the impact of the number of options on the difficulty of *Task 2: Multiple-Choice*. We follow the settings in Section 3 but change the number of options to 2, 3, 5, and 10 respectively. We present the performance of TEXT-DAVINCI-003 and GPT-3.5-TURBO on YAGO in Figure 6. We find that, although a small number of options providing extra context can give the model hints to answer questions, as the number of options increases, the model's performance gradually declines due to the increasing number of distractors.

## 5.7 Generating Triplets vs. Text

We use TEXT-DAVINCI-003 and GPT-3.5-TURBO to directly generate factual triplets about a certain entity (by giving an in-context

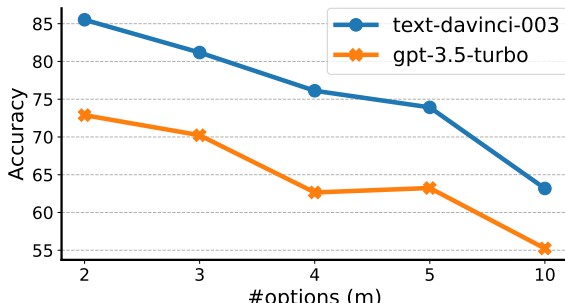

**Figure 6: Impact of the number of answer options on LLM performance. The figure illustrates the performance of TEXT-DAVINCI-003 and GPT-3.5-TURBO on *Task 2: Multiple-Choice* (Multiple-Choice) using YAGO knowledge graph, with varying numbers of answer options (2, 3, 4, 5, and 10). The results show that as the number of options increases, the model's performance declines, indicating that a higher number of distractors makes the task more challenging.**

| Model | Text | | Triplets | |
|---|---|---|---|---|
| | **Precision** | **Recall** | **Precision** | **Recall** |
| DAVINCI | 76.39 | 53.96 | 85.21 | 37.58 |
| TURBO | 77.28 | 57.63 | 91.42 | 37.21 |

**Table 6: Comparison of precision and recall for open-ended text generation and direct triplet generation using TEXT-DAVINCI-003 and GPT-3.5-TURBO. Direct triplet generation results in higher precision but lower recall than open-ended generation.**

example) and reported the precision and recall in Table 6. It can be observed that although the precision has improved, the recall has dropped significantly. We analyzed that this is due to the model generating only a few high-confidence triplets when directly asked for triplets, which led to the aforementioned results. However, for other smaller-scale models, directly generating factual triplets is not feasible, as they cannot adequately understand the prompt's instructions, resulting in poor performance.

## 6 RELATED WORK

*LLM Knowledge Probing.* Research into what knowledge is stored in LLMs has drawn significant interest. Pioneering work like LAMA [48], TempLAMA [12], MMLU [21] quantitatively measured the factual knowledge in these models. Other approaches have expanded these probing techniques, exploring topics like few-shot learning and 2-hop relational knowledge [20]. Furthermore, open-domain question-answering benchmarks like Natural Questions [29], and TriviaQA [25] have been used to measure the practical knowledge abilities of these models, aligning the probing tasks with real-world applications.

*Improving LLM Knowledge Abilities.* Efforts to enhance LLM's knowledge abilities include augmenting language models with KGs for structured, factual knowledge [42, 49] and using retrieval-augmented

methods like RAG [30], REALM [19], and REPLUG [51] to incorporate external documents as a dynamic knowledge source. Further, REMEDI [23] aims to create a finer control over knowledge in LLMs by understanding fact encodings in the model's internal representation system. In parallel, the framework CooK [15] suggests using specialized language models to provide modular and up-to-date knowledge in a collaborative process.

*Extracting Knowledge from LLMs.* The extraction of knowledge from LLMs has become an emerging topic in the research community. Some works focus on constructing KGs from the LLMs [11, 59]. For example, Crawling Robots [11] uses a robot role-play setting to extract named entities and relations by encoding them into actions. Other works utilize the prompt-based paradigm, where they generate knowledge probes in the form of structured prompts [35, 65]. These tools aim to extract and organize the knowledge within an LLM in a human-readable and interpretable way. Furthermore, other techniques involve augmenting training data with recitation tasks to express internally represented knowledge explicitly [54].

*Investigating the Limitation of LLM Knowledge Abilities.* As LLMs have shown promise in knowledge-based tasks, researchers have also started examining the limitations of these models' knowledge abilities. This includes their ability to handle conflicted information [8, 61], recall abilities [39], and self-evaluating skills [27]. By investigating these limitations, researchers aim to not only devise ways to address them but also shed light on how LLMs can operate more effectively in more sophisticated tasks, particularly in professional domains [41, 55].

In summary, while considerable work has been done in probing the knowledge abilities of LLMs, improving these abilities, extracting knowledge, and investigating their limitations, two major aspects have seen less consideration: knowledge utilization and knowledge breadth. These areas are vital for understanding and evaluating the performance of LLMs in more real-world, complex scenarios. Therefore, this calls for a more comprehensive approach, which our proposed KGQUIZ benchmark aims to address, making strides towards a future where LLMs exhibit robust knowledge abilities applicable to a wider range of domains and utilization contexts.

## 7 CONCLUSION

We propose KGQUIZ, a benchmark for probing the knowledge generalization abilities of Large Language Models (LLMs). Unlike previous work, our benchmark focuses on two often-overlooked aspects: the complexity of knowledge utilization and the breadth of knowledge domains. Our benchmark uses structured information from knowledge graphs (KGs) across three diverse domains, and it consists of several tasks representing increasingly complex forms of knowledge utilization. Our experimental results illustrate varying performances of several LLMs across different domains and tasks, underscoring the multi-faceted nature of knowledge abilities in LLMs. This also demonstrates the importance of considering Knowledge Utilization and Knowledge Breadth. We envision KGQUIZ as a comprehensive testbed to evaluate, understand, and improve the knowledge abilities of LLMs across varying domains and tasks.

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

## A  LIMITATIONS

*LM and KG selection.* Due to computational and budget constraints, we restricted our study to ten representative LLMs and three

knowledge graphs each from a different domain. As we plan to make KGQUIZ publicly accessible, further investigation into the performance of a broader range of LLMs on assorted knowledge graphs is left for future endeavors.

*Evaluation Metrics.* Being the case that LLMs might not fully adhere to the context in our prompts, we were required to deploy human-crafted string-processing functions to preprocess the content the models generated, to evaluate the results. This step is susceptible to errors that may lead to inaccurate results. Additionally, the Semantic Match method we utilized is also not without error. Two semantically similar entities could have wildly different referents, which could lead to assessment errors. Addressing the issue of fuzzy match (semantic match) is a direction for future research.

*Knowledge Coverage.* Due to the vast scale of real-world knowledge, we are unable to evaluate whether all the content generated by the model is completely factual in our benchmark. We can only assess whether the content generated by the model aligns with the knowledge stored in the knowledge graphs. However, the coverage of real-world knowledge by the knowledge graph is limited, leading to potential errors in our evaluation. However, as our benchmark is scalable, we can mitigate this limitation to some extent by generating corresponding tasks (questions) using broader (or more applicable) and more up-to-date knowledge graphs.

*Knowledge Breadth.* Our benchmark takes into account the knowledge of three domains: commonsense, encyclopedic, and biomedical. The first two domains are more general, while only biomedical is domain-specific. However, our benchmark can be easily extended to knowledge graphs in other domains, as long as there are corresponding triplet data. This, to some extent, mitigates this limitation.

*KG quality.* Many knowledge graphs contain errors and noise, or outdated knowledge, especially for encyclopedic knowledge graphs like YAGO, which may affect the the validity of our evaluation.

*Prompt Effectiveness.* The prompts we utilized for each question may not necessarily be the most effective. Given the constraints of our budget, we were unable to execute extensive testing on all plausible prompts. Therefore, for *Task 1: True-or-False*, *Task 2: Multiple-Choice Task 4: Factual Editing*, we chose the method of incorporating one in-context example to aid model understanding of the task instructions.

## B  ETHICS STATEMENT

*Privacy.* As KGs encompass a wealth of knowledge on a multifarious range of topics, it can include sensitive or private information. The potential for an LLM, that effectively covers and utilizes this knowledge domain, could generate responses disclosing personal details of individuals or organizations. This introduces privacy concerns and reinforces the need for developing privacy-conscious approaches when leveraging and assessing LLMs and KGs.

*Accessibility.* In making KGQUIZ publicly accessible, we aspire to propel further research on LLMs' knowledge abilities. However, the use of this benchmark may necessitate significant resources due to the inherent complexities of large language models. Similarly, evaluating black-box LLMs could incur significant costs, potentially creating barriers to access to the benchmark for researchers with limited computational resources or budget, contributing to elevated entry barriers in this field.

## C  DISCUSSION

*Performance of LLMs Across Different Knowledge Domains.* Our comprehensive exploration of ten large-scale language models utilizing KGQUIZ revealed that these models exhibited far from uniform performance across diverse knowledge domains and contexts. For instance, the most advanced model, TEXT-DAVINCI-003 displayed varying performance across different knowledge graphs and tasks. Broadly speaking, the performance of this model was the highest on the YAGO knowledge graph, consistently surpassing other models in tasks like true-or-false and multiple-choice. However, when faced with the UMLS knowledge graph representing the biomedical domain, the model showed a significant decline in performance, with ChatGLM and GPT-3.5-TURBO taking the lead instead. These findings emphasize the model's struggles with domain-specific knowledge. Similar trends were also observed with other models like Alpaca, which performed poorly on the multiple-choice task, but displayed a notable improvement on the blank-filling task. Such performance variations across knowledge domains serve as an interesting direction for future research, aiming to investigate the reasons behind such contrasts in LLM performance across diverse knowledge realms.

*LLM Performance Across Knowledge Utilization Contexts.* KGQUIZ has laid emphasis on knowledge utilization patterns along with knowledge domains, providing a comprehensive overview of the knowledge abilities of LLMs. This has enabled a detailed analysis of the models' performance across different knowledge-intensive tasks. A fascinating observation is the influence of task complexity and format on model performance. Alpaca exhibited a significant improvement from *Task 1: True-or-False* to *Task 2: Multiple-Choice*, while the performance of models like TEXT-CURIE-001 dipped. This pattern suggests various models adapt differently to varying complexity and the nature of knowledge utilization at hand. Such insights could be valuable to refine LLM's understanding and handling of tasks, thus warranting further exploration.

*Provide Comprehensive Insight for LLM Evaluation and Comparison.* KGQUIZ is specifically designed to offer a rich set of metrics and contexts for in-depth evaluation and comparison of LLMs' performance across various knowledge domains and utilization contexts. By presenting a fine-grained and multi-perspective analysis, KGQUIZ contributes to a thorough understanding of the strengths and weaknesses of individual LLMs. This not only enables researchers and users to make informed decisions when selecting the best-suited model for a specific task, but also paves the way for the evidence-based development of more capable and versatile LLMs in the future.

*Guidance for Future Development of LLMs.* The performance heterogeneity of LLMs that we observed across varied tasks indicates the challenges certain tasks pose for these models. For instance, LLMs, despite their robust performance on simpler tasks such as True-or-False, struggle to meet the challenge of the increasing complexity of tasks like Factual Editing, emphasizing their limitations

in context-rich, advanced knowledge reasoning. Moving forward, these observations can provide valuable insights for future advancements in the field. Identifying specific areas that require attention and improvement can guide developers to iteratively refine model architectures, enrich training data, and adopt more effective pre-training and fine-tuning methods.

## D    KGQUIZ DETAILS

*In-Context Examples.* Through experiments, we discovered that for the majority of LLMs, their performance in a zero-shot setting is unusually low on some tasks. We think this is because they are unable to precisely comprehend the question's meaning (instructions), and they cannot produce output in the format we expect. Therefore, to preserve fairness without compromise, we have incorporated an in-context example into the prompts of each question for *Task 1: True-or-False*, *Task 2: Multiple-Choice*, and *Task 4: Factual Editing*, which will enable a better assessment of the model's knowledge abilities.

*Threshold for Semantic Match.* For three knowledge graphs, we randomly selected 1,000 entities each. For each entity, we prompted GPT-4 to generate five entities with the same reference and five entities with different references. As a result, we obtained a total of $3 \times 1,000 \times 5$ positive samples and $3 \times 1,000 \times 5$ negative samples. For each sample pair, we calculated their AdaScore. We chose a threshold so that if a positive sample's AdaScore is above the threshold or a negative sample's AdaScore is below the threshold, the sample pair is correctly classified; otherwise, it is misclassified. We selected the threshold that minimized the number of misclassified samples as the Semantic Match threshold.

*LLM-based Triplets Extraction.* We find that it is difficult to measure the similarity between a piece of text and a set of triples. However, evaluating the similarity between two sets of triplets is much easier. So in KGQUIZ Benchmark, we prompt a GPT-3.5 LLM to turn the given model output in natural language into a set of fact triplets. In order to make the model understand the instruction better, we adopt the one-shot setting, as shown in Table 11. To obtain these in-context examples, we first randomly sample k entities from the knowledge graph and find all triples with these entities as head entities. We prompt the TEXT-DAVINCI-003 model to generate a text describing these triples, as shown in Table 10. In this way, we obtain k triple-text pairs as in-context examples. To verify the reliability of this method, we manually evaluate 20 (essay, triplets) pairs. (essay: the TEXT-DAVINCI-003's output text; triplets: the extracted triplets from the model output with our method.) In our human evaluation, the triplets extracted by this method have a precision of 0.87 and a recall of 0.86, demonstrating that our approach has high reliability. The problem with this method is that it extracts triples that do not have the target entity as the head, and the extracted triples do not conform to the format. We expect that providing more in-context examples can help alleviate these issues.

## E    ANALYSIS (CONT.)
### E.1    Knowledge Gap between LLMs and KGs

We conduct qualitative analysis on *Task 5: Open-Ended Text Generation* model outputs and present GPT-3.5-TURBO's generated results

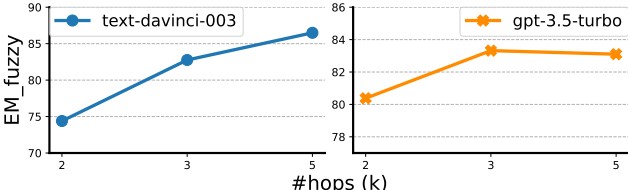

**Figure 7: Effect of the number of hops on LLM performance in the Factual Editing task. The figure shows the Semantic Match scores for TEXT-DAVINCI-003 and GPT-3.5-TURBO on 2-hop, 3-hop, and 5-hop questions generated from YAGO KG. As the number of hops increases, the performance of TEXT-DAVINCI-003 improves, while the performance of GPT-3.5-TURBO exhibits a mixed pattern, indicating that the impact of the hop count on LLM performance varies depending on the model.**

and gold standard answers in Table 8. GPT-3.5-TURBO generated a total of 19 knowledge statements, of which 9 can be matched with triplets in YAGO. Among the remaining 10 knowledge statements that cannot be matched to YAGO, 8 of them are also found to be correct after manual annotation. This indicates that there is a knowledge gap between the parametric knowledge of LLMs and the structured knowledge of KGs. This also further emphasizes the necessity of considering knowledge utilization when discussing the role of KGs in augmenting LLMs. If general information about an entity is what we need, LLMs could provide mostly correct and factual answers; if LLMs need to perform tasks with the exact information in KGs, KG-augmented approaches could still be effective.

### E.2    Number of Hops

*Task 4: Factual Editing* investigates whether LLMs can correct factual mistakes in multi-hop knowledge reasoning chains. We additionally investigate whether the number of hops would affect the difficulty of the factual editing task. We generate 2-hop, 3-hop and 5-hop questions with triplets in YAGO and present the performance of textsctext-davinci-003 and GPT-3.5-TURBO, shown in Figure 7. We observe that as the number of hops increases, the performance of textsctext-davinci-003 improves, with the highest Semantic Match score (86.49) at 5 hops. This indicates that additional context from more hops can be beneficial in identifying and correcting factual inconsistencies in knowledge statements for this model. For GPT-3.5-TURBO, When the number of hops increases from 2 to 3, the performance of the model improves significantly. However, when the number of hops increases to 5, the performance of the model declines slightly but is still higher than that of 2 hops. This once again confirms that the impact of additional context from more hops on LLM performance in the factual editing task depends on the model.

### E.3    Consistency Study

In Section 5.2, we investigate the robustness towards minor changes in prompts and knowledge statements. We present the five different prompts we used in Table 9.

## E.4    Validity of Semantic Similarity Method

In section 2.1, we proposed the Semantic Similarity method for negative sampling. To reduce the computational cost, we only compare similarities among randomly selected m entities. Table 7 presents four *Task 2: Multiple-Choice* questions generated through the ss algorithm sampling. From this, we can see that although there are a few negative sample entities that are not semantically similar to the ground truth entities, most of the negative sample entities have a high semantic similarity to the corresponding ground truth. This demonstrates that this sampling method can, to some extent, select semantically similar entities as negative samples, thereby increasing the difficulty of the problem compared to random sampling.

Owen Pickard is affiliated to [MASK].
A. F.C. Lixa       B. Bideford A.F.C.       C. Stenhousemuir F.C.       D. Erith & Belvedere F.C.
Please choose one from A, B, C, D:

Ground Truth:    B. Bideford A.F.C.

Los Angeles International Airport is connected to [MASK].
A. Guangzhou Baiyun International Airport       B. Honolulu International Airport       C. Rohtak       D. General Rodolfo Sánchez Taboada International Airport
Please choose one from A, B, C, D:

Ground Truth: A. Guangzhou Baiyun International Airport

Nicolás Lodeiro plays for [MASK].
A. Brentwood Town F.C.       B. Club Nacional de Football       C. Thailand national under-23 football team       D. Luverdense Esporte Clube
Please choose one from A, B, C, D:

Ground Truth:       B. Club Nacional de Football

French Polynesia has capital [MASK].
A. Preveza       B. Alberto Lattuada       C. Ulcinj       D. Papeete
Please choose one from A, B, C, D:

Ground Truth:       D. Papeete

**Table 7: Examples of multiple-choice questions generated using the Semantic Similarity (SS) method for negative sampling. The ground truth answer is indicated for each question. Despite a few dissimilar entities, most of the negative samples have high semantic similarity with the ground truth entity, demonstrating the effectiveness of this method**

| Head | Gold | Matched | Factual | Unfactual |
|---|---|---|---|---|
| Mike Judge | {created, King of the Hill} {was born in, Guayaquil} {graduated from, University of California, San Diego} {directed, Office Space} {directed, Idiocracy} {directed, Extract (film)} {created, Office Space} {created, Idiocracy} {created, Extract (film)} {acted in, Office Space} {has gender, male} {lives in, Austin, Texas} | {creates, King of the Hill} {was born in, Guayaquil} {graduated from, University of California} {directs, Office Space} {directs, Idiocracy} {directs, Extract} {produces, Office Space} {produces, Idiocracy} {produces, Extract} | {creates, Beavis and Butt-Head} {creates, The Goode Family} {grew up in, New Mexico} {worked for, tech companies in Silicon Valley} {created, Frog Baseball} {won prize, Primetime Emmy Award} {won prize, Annie Award} {is known for, dry and satirical humor} | {started career as, programmer} {won prize, Peabody Award} |
| John Howard Northrop | {'was born in', 'Yonkers, New York'} {'graduated from', 'Columbia University'} {'works at', 'Rockefeller University'} {'has won prize', 'Nobel Prize in Chemistry'} {'died in', 'Wickenburg, Arizona'} {'works at', 'University of California, Berkeley'} {'has won prize', 'Daniel Giraud Elliot Medal'} {'has academic advisor', 'Thomas Hunt Morgan'} {'has won prize', 'National Medal of Science'} {'has gender', 'male'} {'is citizen of', 'United States'} | {'was born in', 'Yonkers'} {'earned a degree from', 'Columbia University'} {'worked at', 'Rockefeller Institute for Medical Research'} {'won the Nobel Prize in Chemistry in', '1946'} {'passed away in', 'Wickenburg'} | {'was a', 'biochemist'} {'shared the Nobel Prize with', 'James Sumner and Wendell Stanley'} {'worked on', 'isolation and crystallization of enzymes'} {'helped establish biochemistry as', 'a science'} {'conducted research on', 'enzymes'} | {'earned a PhD from', 'University of California'} |

**Table 8: Comparison between the generated answers by the GPT-3.5-TURBO model and the gold standard answers from the YAGO knowledge graph. The matched and factual columns indicate how well the model's answers align with the ground truth and also highlight the factual answers not present in the knowledge graph, reflecting the knowledge gap between LLMs and KGs. The unfactual column shows model-generated answers that are not accurate.**

| ID | Prompt |
|---|---|
| 1 | Is the statement "*[Insert statement here]*" True or False? |
| 2 | Given the statement "*[Insert statement here]*", is this factually correct? Please answer with True or False. |
| 3 | Assess the validity of this claim: "*[Insert statement here]*". Respond with only True or False. |
| 4 | Is the following statement factually accurate? "*[Insert statement here]*" Provide your answer as either True or False. |
| 5 | Can you confirm if this statement is true or false? "*[Insert statement here]*". Reply with just True or False. |

**Table 9: Five prompt templates we used to investigate the robustness towards minor changes in prompts and knowledge statements. We use the sampled knowledge statement to replace *[Insert statement here]* in each template and obtain 5 different prompts for the same knowledge statement.**

Exhaustively express the information from the sentence in a form of subject, relation, object triplets. Triplets should cover all the information from the text, but no more.

Triplets:
Raymond Massey, is married to, Anna Massey
Raymond Massey, acted in, Hotel Berlin
Raymond Massey, acted in, Things to Come
Raymond Massey, was born in, Toronto
Raymond Massey, is married to, Daniel Massey (actor)
Raymond Massey, is affiliated to, Republican Party (United States)
Raymond Massey, acted in, Mackenna's Gold
Raymond Massey, acted in, Abe Lincoln in Illinois (film)
Raymond Massey, has gender, male
Raymond Massey, acted in, The Drum (1938 film)
Raymond Massey, acted in, The Fountainhead (film)
Raymond Massey, acted in, East of Eden (film)
Raymond Massey, acted in, 49th Parallel (film)
Raymond Massey, died in, Los Angeles
Raymond Massey, acted in, The Great Impostor
Raymond Massey, acted in, Mourning Becomes Electra (film)
Raymond Massey, has child, Anna Massey

Text:

**Table 10: An example demonstrating the process used to convert a set of fact triplets about a specific entity into a descriptive text.**

Exhaustively express the information from the sentence in a form of subject, relation, object triplets. Triplets should cover all the information from the text, but no more.

Text:
Raymond Massey, a male actor born in Toronto, was married to Anna Massey and Daniel Massey (actor). He was affiliated to the Republican Party (United States) and acted in numerous films, such as Hotel Berlin, Things to Come, Mackenna's Gold, Abe Lincoln in Illinois (film), The Drum (1938 film), The Fountainhead (film), East of Eden (film), 49th Parallel (film), The Great Impostor, and Mourning Becomes Electra (film). He also had a child with Anna Massey. Raymond Massey died in Los Angeles.

Triplets:
Raymond Massey, is married to, Anna Massey
Raymond Massey, acted in, Hotel Berlin
Raymond Massey, acted in, Things to Come
Raymond Massey, was born in, Toronto
Raymond Massey, is married to, Daniel Massey (actor)
Raymond Massey, is affiliated to, Republican Party (United States)
Raymond Massey, acted in, Mackenna's Gold
Raymond Massey, acted in, Abe Lincoln in Illinois (film)
Raymond Massey, has gender, male
Raymond Massey, acted in, The Drum (1938 film)
Raymond Massey, acted in, The Fountainhead (film)
Raymond Massey, acted in, East of Eden (film)
Raymond Massey, acted in, 49th Parallel (film)
Raymond Massey, died in, Los Angeles
Raymond Massey, acted in, The Great Impostor
Raymond Massey, acted in, Mourning Becomes Electra (film)
Raymond Massey, has child, Anna Massey

Exhaustively express the information from the sentence in a form of subject, relation, object triplets. Triplets should cover all the information from the text, but no more.

Text:
<Model Response of Task 5: Open-Ended Text Generation>

Triplets:

**Table 11: An example prompt for the GPT-3.5 LLM to extract information triplets from the model's open-ended text generation response.3**

