# OpenReview forum: "KGQuiz: Evaluating the Generalization of Encoded Knowledge in Large Language Models"
_ACM.org/TheWebConf/2024/Conference — TheWebConf24 Oral_

### Official Review · Reviewer_gQsK · 2023-11-23

**Novelty:** 3
**Technical Quality:** 4

**Review:**

The paper is a good contribution to the field of LLM research, offering a comprehensive benchmark for evaluating LLMs across a range of tasks and domains. Its methodology is robust, and the results are presented with clarity and detail.
Pros:
1. The introduction of KGQUIZ, a scalable framework for evaluating LLMs, is good. It assesses LLMs across five knowledge-intensive tasks and three knowledge domains, offering a thorough understanding of LLMs' abilities.
2. The paper conducts extensive experiments with 10 different LLMs, providing a broad perspective on their performance in various tasks and domains.
3.  The tasks vary in complexity, ranging from true-or-false to open-ended text generation, offering insights into the strengths and limitations of LLMs in different contexts.

Cons:
1. Domain Limitation: Although the study covers three domains, its focus on biomedical knowledge as the only representative of domain-specific knowledge may limit the broader applicability of its findings to other specialized fields. This seems contradictory to the paper's mention of "covering diverse knowledge sources, subject areas" as it appears to be limited to the field of biology only.
2. Task Design and Flexibility: The proposed tasks are quite rigid in form, almost equivalent to knowledge graph triplets, and lack direct multi-step reasoning tasks. This approach might not fully capture the nuanced capabilities of LLMs, such as emergent abilities like contextual learning and the chain of thought ability.
3. Task Weighting: Assigning equal weight to all tasks may not accurately represent their relative importance or complexity in real-world applications.
4. Lack of Model Descriptions: The absence of detailed introductions for the 10 open-source and black-box LLMs, particularly the differences between models like ADA, BABBAGE, CURIE and DAVINCI is a significant oversight for readers in understanding the experimental methods and analyzing results.
5. Lack of Deep Analysis: While the analysis is comprehensive, it falls short in deeply exploring why specific models like TURBO or DAVINCI excel in certain tasks or domains. Also, there is a lack of fundamental explanation for the experimental observation that the same large model varies greatly across different knowledge domains.
6. Ambiguous Experimental Conclusions: The paper primarily focuses on introducing a new benchmark and conducting experiments. However, it falls short of offering substantive research conclusions, leaving the reader with an incomplete understanding of the broader implications of the findings.

**Questions:**

1. You've chosen commonsense, encyclopedic, and biomedical domains for your study. Can you explain through more diverse data analysis if these three domains are comprehensive enough to fully understand the knowledge capabilities of LLMs? Could you elaborate on how these particular domains were chosen and whether including additional or different domains like mathematics and coding subjects might alter the study's conclusions?
2. Your study mainly employs the Semantic Similarity strategy for negative sampling. Could you discuss the potential impact of the other three negative sampling strategies on the outcomes? How might they have challenged the LLMs differently?
3. In your methodology, all tasks are seemingly weighted equally. Could you comment on how different weightings, reflecting the relative importance or real-world applicability of these tasks, might affect the study's conclusions?
4. Your results show considerable performance variability across different LLMs and tasks. Could you provide more insights into why certain models excel in specific tasks or domains? And why do certain models show significant variations in performance across different tasks or domains? Is it related to their training data, architecture, or some other factors?
5. What practical significance does your work and experimental conclusions have for the development of LLMs? What are the key areas you would recommend focusing on for the future development of LLMs? Are there specific weaknesses or gaps that emerged as particularly crucial to address?
6. Regarding expanding knowledge domains and compared to other benchmarks, what advantages does the KGQUIZ benchmark have? Which new domains do you plan to expand into? How do you envision the KGQUIZ benchmark evolving with the rapid advancements in LLMs?

**Ethics Review Description:**

No ethics issues

**Reviewer Confidence:**

4: The reviewer is certain that the evaluation is correct and very familiar with the relevant literature

**Scope:**

3: The work is somewhat relevant to the Web and to the track, and is of narrow interest to a sub-community

---

### Official Review · Reviewer_nzWA · 2023-11-24

**Novelty:** 6
**Technical Quality:** 6

**Review:**

# Review of KGQUIZ: Evaluating the Generalization of Encoded Knowledge in Large Language Models"
- This paper addresses the critical need for improved evaluation mechanisms for large language models, extending beyond traditional task-specific benchmarks. It categorizes tasks into varying complexity levels, ranging from simple binary true/false questions to multiple choice, blank filling, factual editing, and open-ended text generation.
It provides a detailed description of the makeup and structure of its approach to benchmarking LLM performance in a broad sense, evaluating various models and providing a comparative perspective. In doing so, it identifies gaps and varying focuses among different language models.
- In the opinion of the reviewer, the paper provides a useful contribution to a key issue in the field of language model research. It should be of interest to the conference audience.

## Pros
- The paper tackles the significant problem of evaluating large language models, which is fundamental for understanding their capabilities and limitations.
- It introduces a systematic approach, categorizing tasks based on complexity and difficulty, which provides a more nuanced understanding of model performance. The reviewer found the description of the approach to be thorough and illuminating.
- The paper effectively identifies and discusses the differences and gaps between various language models, shedding light on their distinct focuses and capabilities.
- By focusing on evaluation techniques, the paper contributes valuable insights to the ongoing research and development in the field of language modeling.

## Cons
- The chosen tasks for evaluation, although varied, may not encompass all aspects of language model capabilities, possibly leading to a partial understanding.

**Questions:**

- One of the challenges in applying LLMs is their tendency to overproduce on extraction tasks, i.e. they generate triples that are not in the benchmark. Half of the problem there is what is termed hallucination; i.e. triples that are incorrect and not supported by the evidence. The other half of the problem are triples that are factually correct, but due to their absence in the benchmark are treated as incorrect. Using an approach where the benchmark is the gold standard, the latter case is one where the LLM's extraction will be considered an error when in fact it is not. How would your approach mitigate the latter problem?

**Reviewer Confidence:**

3: The reviewer is confident but not certain that the evaluation is correct

**Scope:**

4: The work is relevant to the Web and to the track, and is of broad interest to the community

---

### Official Review · Reviewer_pd1P · 2023-11-24

**Novelty:** 6
**Technical Quality:** 7

**Review:**

The paper introduces KGQUIZ, a knowledge-intensive benchmark designed to systematically evaluate the knowledge generalization abilities of Large Language Models (LLMs). The benchmark, constructed from triplet-based knowledge, encompasses three knowledge domains and includes tasks of increasing complexity, such as true-or-false, multiple-choice QA, blank filling, factual editing, and open-ended knowledge generation. The evaluation of 10 LLMs on the KGQUIZ benchmark reveals the performance in straightforward knowledge QA tasks, while more complex reasoning and domain-specific facts pose significant challenges, positioning KGQUIZ as an interesting testbed for analyzing nuanced variations in LLMs' performance across diverse domains and task formats.
**Pros:**
* The paper includes both open and closed model language
* The paper is "self contained", the task is clear and the results are not oversold, the focus is on the benchmark and not on "the LLMs reasoning capabilities"
* The pipeline is well explained ad documented

**Cons:**
* It is not clear if the "open" models are open in their code or also in the train set. It is not detrimental to the work per se, but it would be interesting to have in the Appendix the information that we are certain that some models had in their training corpus the knowledge graphs used for this work
* The prompts are not provided. Could you attach them in a separate repository?
* On page 11 in the Appendix there is a line which is mildly overflows
* it is "planned" to be made available as an open benchmark, but although I understand the author's concerns, even at the actual state it would be good to release the pipeline, and I would ask you to make it open as a necessary condition of acceptance of the paper to this venue.

**Questions:**

Please see above.

**Ethics Review Description:**

\

**Reviewer Confidence:**

3: The reviewer is confident but not certain that the evaluation is correct

**Scope:**

4: The work is relevant to the Web and to the track, and is of broad interest to the community

---

### Official Review · Reviewer_EeRH · 2023-11-27

**Novelty:** 5
**Technical Quality:** 5

**Review:**

**Summary:**

The paper presents KGQuiz, a new benchmark to investigate the encoded knowledge and reasoning capabilities of a variety of large language models. The paper highlights the limitations of existing probing, QA, and reasoning benchmark datasets and presents a dataset built from YAGO, ConceptNet, and UMLS for the following five tasks: 1) true or false questions, 2) MPC questions, 3) blank-filling, 4) factual editing, and 5) text generation.


**Review:**

I like the topic and the idea of creating a new benchmark dataset for probing LLMs, however, the concrete selling point of this work is not completely clear to me. The paper is motivated by the fact that existing benchmarks do not look at knowledge utilization and do not cover multiple domains. Isn’t KILT at least doing the first of the two? And was the addition of different domains that valuable in the end? Looking at the results in your work, the performance among domains is often very similar. Probably because you mostly probed for knowledge about long-tail entities.
Overall, I am not sure, if this new benchmark is needed and adds much value over existing benchmarks. Also, some details in the benchmark construction, e.g. random entity sampling, and the semantic matching metric, seem not optimal in my opinion.


**Details:**

-	One problem I see is the similarity to the KILT benchmark dataset from Meta that was published a couple of years ago. I see that the paper is mentioned in the introduction of your work, however, I see a large overlap in the tasks of the two works and therefore would encourage the authors to explicitly discuss the novelty of their own work regarding KILT.
-	Furthermore, I doubt that using randomly sampled entities from the knowledge graphs is a good choice for building datasets for these tasks. In my experience, this will result in a dataset only consisting of long-tail entities. I am not sure if this is what you want to evaluate in the end. As demonstrated by your results all models, but the huge ones have a performance around random chance. Hence, they don’t know anything about any of the entities that are in your dataset.
-	The difference between the exact match metric and the semantic similarity metric is huge. I see that there are difficulties with EM for this task, but semantic similarity might also be likely to evaluate wrong results as correct. I think further analysis is needed here to figure out how to evaluate these results.
-	Section 5 is kind of unorganized and feels like a collection of analysis and results that did not fit anywhere else. I think it would be nice to add some explanatory and motivating texts for the different experiments you present here.

**Strengths:**

-	Probing LLMs in knowledge-intensive tasks is an important and interesting problem.
-	The paper is well-written, uses nice images to explain the different tasks and is a nice read.
-	The number of experiments and models tested in this work is large.
-	The paper has an extensive list of related works.

**Weaknesses:**

-	This paper does not present any methodological novelty, but “only” presents a new benchmark dataset for probing LLMs.
-	The similarity to KILT seems to be rather large.
-	A motivation for a selection of exactly these 5 tasks is unclear to me.
-	Random entity samples from a KG might not be the best choice since this most likely leads to long-tail entities being in your dataset. Hence, the low performance of most models.

**Questions:**

-	I wonder if your negative sampling strategies might actually lead to some false negative examples. It is commonly known that KGs are rather incomplete. Have you performed any analysis on this or have any other insight that you can share with me about this?
-	I am unsure why factual editing is a relevant task here. Why is this substantially different than the other tasks?
-	I am not sure I understand how you computed the average ranks in Table 1. I guess it is either the average rank over the datasets or the tasks respectively. Why exactly are they relevant and you do not just present the performance metric for each task instead? Those would implicitly contain the rank of the model, or not?

**Reviewer Confidence:**

4: The reviewer is certain that the evaluation is correct and very familiar with the relevant literature

**Scope:**

3: The work is somewhat relevant to the Web and to the track, and is of narrow interest to a sub-community

---

### Decision · Program_Chairs · 2024-01-22

**Decision:**

Accept (Oral)

**Comment:**

* (+) There are no concerns with scope
 * (+) Novelty is unambiguous
 * (+) Concerns with technical quality have been addressed in reviewer/author discussion